# Recessive Transition Mechanism of Arable Land Use Based on the Perspective of Coupling Coordination of Input–Output: A Case Study of 31 Provinces in China

Yi Lou, Guanyi Yin *, Yue Xin, Shuai Xie, Guanghao Li, Shuang Liu and Xiaoming Wang

College of Geography and Environment, Shandong Normal University, Jinan 250000, China;
2019020678@stu.sdnu.edu.cn (Y.L.); 2018020745@stu.sdnu.edu.cn (Y.X.); 2020020735@stu.sdnu.edu.cn (S.X.);
201914010224@stu.sdnu.edu.cn (G.L.); 201814010119@stu.sdnu.edu.cn (S.L.); 204020@sdnu.edu.cn (X.W.)
* Correspondence: 616071@sdnu.edu.cn

**Abstract:** In the rapid process of urbanization in China, arable land resources are faced with dual challenges in terms of quantity and quality. Starting with the change in the coupling coordination relationship between the input and output on arable land, this study applies an evaluation model of the degree of coupling coordination between the input and output (D_CCIO) on arable land and deeply analyzes the recessive transition mechanism and internal differences in arable land use modes in 31 provinces on mainland China. The results show that the total amount and the amount per unit area of the input and output on arable land in China have presented different spatio-temporal trends, along with the mismatched movement of the spatial barycenter. Although the D_CCIO on arable land increases slowly as a whole, 31 provinces show different recessive transition mechanisms of arable land use, which is hidden in the internal changes in the input–output structure. The results of this study highlight the different recessive transition patterns of arable land use in different provinces of China, which points to the outlook for higher technical input, optimized planting structure, and the coordination of human-land relationships.

**Keywords:** land use transition; arable land use; input–output; spatio-temporal variation; movement of spatial barycenter; optimization of arable land use

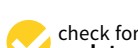



## 1. Introduction

Historically, the world has not witnessed such rapid urbanization as that which has taken place over the past several decades in China [1]. Measured as the proportion of permanent urban population in the total population, the urbanization of China has risen from 17.9% in 1978 (the year of the reform and opening-up in China) to 59.85% in 2018. Along with the urbanization process, China's economic structure is also undergoing rapid transition. The proportion of China's nonagricultural economy (the ratio of the GDP of the secondary and tertiary industries in the total national GDP) increased from 72% in 1978 to 93% in 2018, showing a significant shift in economic development focus towards nonagricultural industries. In this vast wave, the use of arable land is facing various challenges. On the one hand, accompanied by a decrease in the rural population (from 790.14 million in 1978 to 56.401 million in 2018), the urban population increased from 172.45 million in 1978 to 831.37 million in 2018 [2]. By 2018, more than 200 million rural residents had migrated to urban areas, indicating a large transition of farmers' livelihood from traditional agricultural production to employment in secondary and tertiary industries in cities [3,4]. On the other hand, along with a rapid expansion of construction land area (from 5845 hm$^2$ in 1978 to 56,075.9 hm$^2$ in 2018, with an annual growth rate of 21.5%), the urban built-up area in China increased synchronously from 6720.5 to 58,455.7 hm$^2$, with an annual increase rate of 19.2%. During the same period, the annual growth rate of the urbanization rate in China was 1.04%. This striking development pattern indicated that the "urbanization of land"

is much faster than the "urbanization of population," which has posed high pressure on the loss of arable land resources [5]. From 2004 and 2018, 10,693.20 hm$^2$ of arable land was converted into construction land in China. It is undeniable that the development of urbanization in China emphasizes the limitation of arable land resources and stimulates arable land intensity. However, it also brings more dual challenges of both the agricultural labor force and arable land resources [6].

In recent years, the question of how to effectively protect arable land and ensure food security has been a hot topic for policymakers and researchers. Since 1998, the Chinese government has implemented a series of arable land protection policies. In China's land use constraint system, users are required to use the land strictly in accordance with the prescribed land uses and emphasize that the transition of arable land to construction land must be examined and approved by the higher government. In the balanced system of requisition–compensation for arable land, local governments are required to complement the same area of high-quality arable land if construction occupies arable land. In the stripping and reuse system of tillage soil, users of construction land who occupied high-quality arable land are required to strip the topsoil of arable land and move the fertile soil to other arable lands for further reuse. In the compensation system of arable land protection, the rural collective economic organization and the farmers who implement farmland protection can obtain an annual subsidy of 6.3–20 RMB/hm$^2$, which varies in different regions. Every year, the No.1 document of the State Council of China restates the red line of $1.2 \times 10^8$ hm$^2$ of arable land, highlights the stipulation of arable land rotation and land fallow, emphasizes the implementation of a permanent basic farmland protection system, and keeps improving the balance system of requisition–compensation of arable land. A series of policies have curbed the rapid decline in arable land area and has realized the coordination and unity of protecting arable land resources to a certain extent [7,8].

However, the demand for land continued to increase with the rapid urbanization process, which caused negative results in the amount and quality of arable land. In the practice of balancing the system of requisition–compensation of arable land, the supplemented arable land tends to be insufficient and low-quality. In some local areas, high-quality arable land is used for construction, and low-quality land is used for supplementing arable land, which gives great uncertainty to the overall efficiency of arable land use [9,10]. In terms of arable land quality, scholars also point out various quality problems in arable land use. The decrease in nutrient content, loss of cultivated layer soil, nonpoint agricultural pollution caused by excess pesticide and fertilizer use are frequently discussed in arable land use [11–15]. From the perspective of the internal complexity of arable land use, the fluctuating multiple cropping index, increasing nongrain planting structure, and arable land abandonment problems have had intricate far-reaching influences on arable land use [16,17]. This phenomenon formed an arable land use mode with the characteristic of "heavy use and light maintenance," which will inevitably lead to the continuous decline of arable land use efficiency in China. For fear of further deterioration, wide observations of arable land use change, and strong measures of arable land use optimization are urgently needed.

Faced with the above problems, the importance and necessity of land use transition have been highlighted. With socioeconomic development, Long et al. proposed that land use transitions will appear in conflicts and transitions among different land use types, which will result in a new balance along with periodic change [18]. Among the various transitions of land use, Chen highlighted that the most typical transition in China is the transition from arable land to construction land, which needs to be optimized to better promote integrated urban-rural development [19]. Lyu et al. believes that sustainable of arable land use requires rational of input structure, improvement of output and greener ecological environment [20]. In this perspective, the change in input–output structure will profoundly regulate sustainable intensified arable land use. More specifically, the arable land use should not blindly increase output by increasing the input, but reasonably coordinate the spatio-temporal relation of input-output matters most. An atypical characteristic of the

input–output change on arable land is the coupling relationship between the specific input system (combining fertilization, sowing, pesticide, and mechanical power) and the output of arable land. This kind of internal change in the coupling relationship between input and output on arable land is vital but invisible, it may constitute the recessive characteristic of arable land use transition. What kind of coupling relationship between the input and output on arable land is presenting? How does the arable land use transition change in different areas? Is there any space for arable land use transition? These questions are becoming meaningful perspectives for arable land use improvement. To our knowledge, previous studies mostly focus on the discussion of arable land quality and quantities and the evaluation of the overall land use efficiency and risk [21–24]. Less attention has been given to the arable land use transformation mechanism from the perspective of input–output change. As a result, the above questions have not yet been fully answered.

To fulfill the research gaps, this study aims to further explore the law of the recessive transition mechanism of arable land use based on the input-output perspective. By evaluating the degree of coupling coordination of input–output on arable land (D_CCIO) in 31 provinces of mainland China, this study raises the following scientific questions:

1.　In the rapid urbanization wave of China, what are the spatiotemporal changes in the input–output of arable land?
2.　Is the input and output of arable land highly coupled and sustainable?
3.　What are the characteristics of arable land use transition among different regions, and what is the enlightenment for the policy of optimizing arable land use mode?

Based on the above research objectives, this paper is divided into the following parts: Section 2 introduces the research area, methods, and data; Section 3 introduces the research results; Section 4 discusses the enlightenment of the research results in depth, and Section 5 summarizes the main conclusions of this paper.

## 2. Research Area and Methods

### 2.1. Research Area and Data Source

This paper selects 31 provincial administrative units of mainland China as the research area (Figure 1). In the research process, the basic data involved in the calculation of each indicator are obtained from the China rural statistical yearbook (2009, 2019) and China population and employment statistical yearbook (2009, 2019). The spatial boundaries of administrative areas are obtained from the Resource and Environment Science and Data Center of the Institute of Geographic Sciences and Natural Resources Research, Chinese Academy of Sciences.

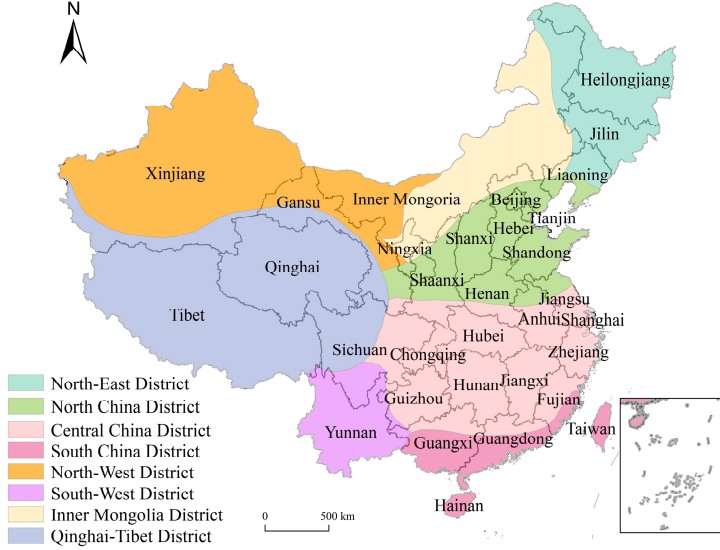

**Figure 1.** The 31 provinces of China in this study.

*2.2. Indicator System of Input–Output Analysis on Arable Land*

This study establishes an evaluating indicator system of the coupling coordination for input–output on arable land. Indicators involved in the evaluation were chosen based on the principles of comprehensiveness, independence, and accessibility. First of all, since 2015, the Ministry of Agriculture in China began to advocate zero growth of the use of fertilizers and pesticides. Because fertilizers and pesticides are essential elements in the growth of crops, therefore, the rational use of them is of vital importance for improving arable land use efficiency [25–27]. Secondly, mulching film can improve the utilization efficiency of arable land and water, which is widely used in areas with water shortages and low temperatures. However, it is important to deeper analyze the change of mulching film use, because the residues are difficult to degrade and will destroy soil pore structure [28]. Finally, as complementary roles of planting, sowing, and harvesting, observing the change of labor force and agro-machinery is necessary for analyzing arable land use [29,30].

Considering the vast territory of China, different climatic and geographical conditions nourish different local crops, so it is necessary to select a variety of crops in the output indicators. Because the total output of grain, oil crops, cotton, sugar crops, tobacco, vegetables, and fruits accounted for more than 95% of the country's crop output, the yields of the above seven types of crops were chosen as the output indicators (Table 1) [21,31–33].

**Table 1.** Indicators of the subsystem of input–output on arable land.

| Subsystem | Content | | Subsystem | Content | |
|---|---|---|---|---|---|
| | Nitrogen fertilizer (kg) | I1 | | Grain (kg) | O1 |
| | Phosphorus fertilizer (kg) | I2 | | Oil crop (kg) | O2 |
| | Potash fertilizer (kg) | I3 | | Cotton (kg) | O3 |
| Input of arable land (I) | Compound fertilizer (kg) | I4 | Output of arable land (O) | Sugar crop (kg) | O4 |
| | Mulching film (kg) | I5 | | Tobacco (kg) | O5 |
| | Mechanical power (kg) | I6 | | Fruit (kg) | O6 |
| | Pesticide (kg) | I7 | | Vegetable (kg) | O7 |
| | Labor (person) | I8 | | | |

*2.3. Evaluation of the Degree of Coupling Coordination Between Input and Output on Arable Land (D_CCIO)*

2.3.1. Quantification of the Input–Output Subsystems on Arable Land

Based on thermodynamic principles, the entropy method has been widely used as an objective method in engineering, social and economic fields. This method uses the information entropy to calculate the entropy value according to the variation degree of each indicator and can effectively solve the problem of information overlap between multiple index variables [34]. After determining the weight of each indicator using the entropy method, we can calculate the indexes of the input–output subsystem using the following process.

(1) Standardization of indicators:

To eliminate the influence of data dimensions and units on the evaluation result, the original data are usually transformed into dimensionless data. The specific formula is as follows:

$$U_{ij} = \frac{P_{ij} - minP_{ij}}{maxP_{ij} - minP_{ij}} \tag{1}$$

where $U_{ij}$ represents the standardized value for the $j$-th indicator of the $i$-th item and $P_{ij}$ is the value for the $j$-th indicator of the $i$-th item.

(2) Calculate the intermediate parameters ($M$):

$$M = \frac{P_{ij}}{\sum_{i=1}^{n} P_{ij}} \tag{2}$$

(3) Calculate the entropy ($e_j$) of the $j$-th indicator:

$$e_j = \frac{-\left(\sum_{i=1}^{n} M_{ij} \times \ln M_{ij}\right)}{\ln n} \tag{3}$$

(4) Calculate the utility value of each index ($d_j$):

$$d_j = 1 - e_j \tag{4}$$

(5) Calculate the index weight ($w_j$):

$$w_j = \frac{d_j}{\sum_{j=1}^{n} d_j} \tag{5}$$

(6) Calculation of each subsystem score:

$$I = \sum_{j=1}^{n} w_j M_{ij} \quad O = \sum_{j=1}^{n} w_j M_{ij} \tag{6}$$

where $I$ and $O$ indicate the input index and output index on arable land, respectively.

2.3.2. Evaluating the Degree of Coupling Coordination of Input and Output (D_CCIO)

To study the degree of coupling coordination for the input–output on arable land, a coupling coordination evaluation model was built as follows [35]:

$$C = 2\sqrt{\frac{I \times O}{(I + O)^2}} \tag{7}$$

where $C$ is the degree of coupling, indicating the interaction intensity of the two-word systems, and $C \in [0, 1]$. The greater the coupling degree is, the stronger the interaction between the two subsystems, and vice versa.

The coupling model is limited in that it cannot reflect the development level of two systems, and "false" high coupling results may appear in two low-level systems. To avoid this problem, the coupling coordination degree model was used to accurately evaluate the coupling coordination relationship between the input and output of arable land at the provincial level.

$$T = \alpha I + \beta O \tag{8}$$

$$D\_CCIO = \sqrt{C \times T} \tag{9}$$

where $T$ is the comprehensive coordinating index of the input and output of arable land, which reflects the effect or contribution of the integrated synergy of the input and output of arable land. Both $\alpha$ and $\beta$ are weights to be determined. The input system and output system of arable land are equally important and reference previous achievements $\alpha = \beta = 0.5$ [34,36]. D_CCIO is the degree of coupling coordination between input and output on arable land, abbreviated as $D\_CCIO$, $D\_CCIO \in [0, 1]$. By referring to relevant references and combining them with the actual situation of the research area, $D\_CCIO$ is divided into sections (Table 2), and quantitative judgment and analysis are made according to the criteria given herein [37].

**Table 2.** Classifications of degree of coupling coordination between input and output on arable land.

| Categories | Development Modes between Sub Systems of Input and Output | Grades of D_CCIO | Classes |
|---|---|---|---|
| Balanced arable land use | I > O: Balanced arable land use with output lagged<br>I = O: Balanced arable land use with input and output synchronized<br>I < O: Balanced arable land use with input lagged | 0.91~1.00<br>0.81~0.90<br>0.71~0.80<br>0.61~0.70 | Extremely balanced arable land use<br>Seriously balanced arable land use<br>Moderately balanced arable land use<br>Slightly balanced arable land use |
| Transitional arable land use | I > O: Transitional arable land use with output lagged<br>I = O: Transitional arable land use with input and output synchronized<br>I < O: Transitional arable land use with input lagged | 0.51~0.60<br>0.41~0.50 | Barely balanced development<br>Barely unbalanced development |
| Unbalanced arable land use | I > O: Unbalanced arable land use with output lagged<br>I = O: Unbalanced arable land use with input and output synchronized<br>I < O: Unbalanced arable land use with input lagged | 0.31~0.40<br>0.2~0.3<br>0.1~0.2<br>0~0.1 | Slightly unbalanced arable land use<br>Moderately unbalanced arable land use<br>Seriously unbalanced arable land use<br>Extremely unbalanced arable land use |



*2.4. Standardized Deviation Ellipse of the Input and Output on Arable Land*

The analysis of the standard deviation ellipse (SDE) was first proposed by Lefever in 1926 and has been used as a feasible tool to describe the directivity of the spatial distribution [38,39]. The spatial and temporal evolution of geographical elements in spatial distribution range, direction, and shape can be described through the variation of parameters such as the barycenter, the size of the major and minor axes, and the standard difference of the major and minor axes of the ellipse to comprehensively reveal the spatial and temporal evolution characteristics and process of regional development from multiple perspectives [40]. The main parameters of the SDE are calculated as follows:

$$\overline{X} = \frac{\sum_{i=1}^{n} x_i}{n}, \ \overline{Y} = \frac{\sum_{i=1}^{n} y_i}{n} \tag{10}$$

$$SDE_x = \sqrt{\frac{\sum_{i=1}^{n}\left(x_i - \overline{X}\right)^2}{n}}, \ SDE_y = \sqrt{\frac{\sum_{i=1}^{n}\left(y_i - \overline{Y}\right)^2}{n}} \tag{11}$$

where $x_i$ and $y_i$ are the coordinates for feature $i$, $\left(\overline{X}, \overline{Y}\right)$ represents the coordinates of the spatial barycenter for the features, and n is equal to the total number of features. $SDE_x$ and $SDE_y$ represent the major and minor axes of the ellipse. By using the SDE method in ArcGIS 10.2, the directivity of the spatial distribution of input and output on arable land can be visualized. The evolution process and law of the spatial distribution of the input and output factors of arable land in the province of China were obtained.

*2.5. Analysis of the Spatial Barycenter of Input and Output on Arable Land*

Barycenter modeling, which has been extensively utilized in the fields of urban planning, economic geography, and land use science, is a preferred modeling approach that traces the spatial movement direction of barycenters for targeted objects. Moreover, movement direction and distance to the center of gravity can reflect changes in quantity and changing trends of the targeted object over time [41]. Differing from the qualitative description of the spatial change in arable land, the law of barycenter migration can represent the whole dynamic evolution process of element distribution. The equation of the movement distance for the barycenter can be expressed as follows:

$$Distance = \sqrt{\left(X_{t2} - X_{t1}\right)^2 + \left(Y_{t2} - Y_{t1}\right)^2} \tag{12}$$

where *Distance* is the movement distance of barycenter(km), and $X_{t1}$, $X_{t2}$, and $Y_{t1}$, $Y_{t2}$ are coordinates of spatial barycenter of different inputs and outputs for the years *t*1 and *t*2.

**3. Result Analysis**

*3.1. The Spatiotemporal Change in the Input of Production Material on Arable Land*

3.1.1. Provincial Input of Production Material

In 2008–2018, three kinds of production material inputs on arable land experienced a significant decrease. The input of nitrogen fertilizer, pesticide, and labor on arable land decreased by $2.37 \times 10^9$ kg, $1.69 \times 10^8$ kg, and $4.80 \times 10^7$ person, respectively. The reduction rate of the above three inputs on arable land fell by more than 10.0%. In contrast, other inputs on arable land showed a rising trend. The most significant one is the decline in compound fertilizer input, which increased from $1.61 \times 1010$ to $2.27 \times 1010$ kg in 2008–2018 (with an increase rate of 40.9%). In addition, the input of plastic film increased from $1.11 \times 10^9$ to $1.40 \times 10^9$ kg, and the input of mechanical power increased from $8.22 \times 10^8$ to $1.00 \times 10^9$ kw, both of which increased by more than 20%. Comparatively, the increase rate of phosphate fertilizer and potash fertilizer was not significant (<10%).

Figures 2 and 3 represent the input factors on arable land in 2008 and 2018, respectively. The subfigures A-H illustrate the input of nitrogen fertilizer, phosphate fertilizer, potash fertilizer, compound fertilizer, pesticide, mulching film, mechanical power and labor, re-

spectively. The production material input on arable land in the Huang-Huai-Hai Plain area is relatively prominent. Various kinds of input on arable land in this area accounted for over 28% of the national total. In particular, mechanical power accounted for as high as 46.97% in 2008. The standardized ellipse shows that the directional distribution of various inputs is of two types: northeast-southwest directional distribution (Figure 2A–H except Figure 2F) and northwest-southeast directional distribution (Figures 2F and 3F). Except for the input of plastic film, various inputs showed directivity along the northeast-southwest direction, which is roughly consistent with the distribution of China's major grain-producing areas. The high-value area of various inputs also coincides with the distribution of the main grain-producing areas, indicating that the production material input is much higher in these areas. Conversely, the plastic film shows a distribution directivity along the northwest–southeast, and the high-value areas are mainly concentrated in Xinjiang, Gansu, Sichuan, and Yunnan provinces. In addition, the proportion of provinces with the top 2 highest inputs of the mulching film increases from 30% to 38%, showing an increasing degree of spatial aggregation for mulching film input.

In terms of specific elements, the polarization characteristic of nitrogen fertilizer input is prominent. Although the total amount of nitrogen fertilizer across the country showed a decreasing trend, the number of provinces with the highest top 2 inputs and the minimum input of nitrogen fertilizer increased by eight and four, respectively. This indicated that the max-min range among different polar means is prominent. Conversely, the total amount of mechanical power across the country increased, but provinces with the highest input and the lowest input both decreased. Similarly, the spatial distribution of pesticide input (Figures 2E and 3E) maintained a "U"-shaped pattern, but the number of provinces with the highest input decreased significantly.

### 3.1.2. Input of Production Material on Per Unit Area

Considering that different arable land areas exist among different provinces, this study also uses the input of production material per unit area to reflect the "intensity" of input. Except for phosphate fertilizer, the change trend of input per unit area and total input of all production materials was consistent. The input of nitrogen fertilizer per unit area decreased from 203.85 to 163.31 kg/hm$^2$. The reduction rates of pesticide and labor force input per unit area were also high (the former 20.05%, the latter 18.57%). Unexpectedly, the increase in the total input of phosphate fertilizer was accompanied by a decrease in input per unit area (from 61.65 to 52.61 kg/hm$^2$). Conversely, the national average input of compound fertilizer per unit area has increased significantly from 139.20 to 171 kg/hm$^2$ (increase rate of 22.85%). It is worth noting that although the increment of potash fertilizer is very low (from 0.23 to 0.28 kg/hm$^2$), its rate of increase is the highest (23.01%). Comparatively, the growth rate of plastic film and mechanical power input per unit area was low (<10%).

Figures 4 and 5 represent the input factors on per unit area of arable land in 2008 and 2018, respectively. The subfigures A–H illustrate the input of nitrogen fertilizer, phosphate fertilizer, potash fertilizer, compound fertilizer, pesticide, mulching film, mechanical power and labor on per unit area of arable land, respectively. The spatial distribution of production material inputs per unit area is significantly different from that of total inputs. Unlike the northeast–southwest directivity of the standardized ellipse (Figures 2 and 3), the input of nitrogen fertilizer, phosphorus fertilizer, mechanical power, and labor force in the unit area did not show obvious spatial directivity. A possible reason is that the input of pesticides and fertilizers per unit area in the main grain-producing areas is no longer significantly higher than that in the nonmain grain-producing areas (the former 88.30 kg/hm$^2$, the latter 77.68 kg/hm$^2$).

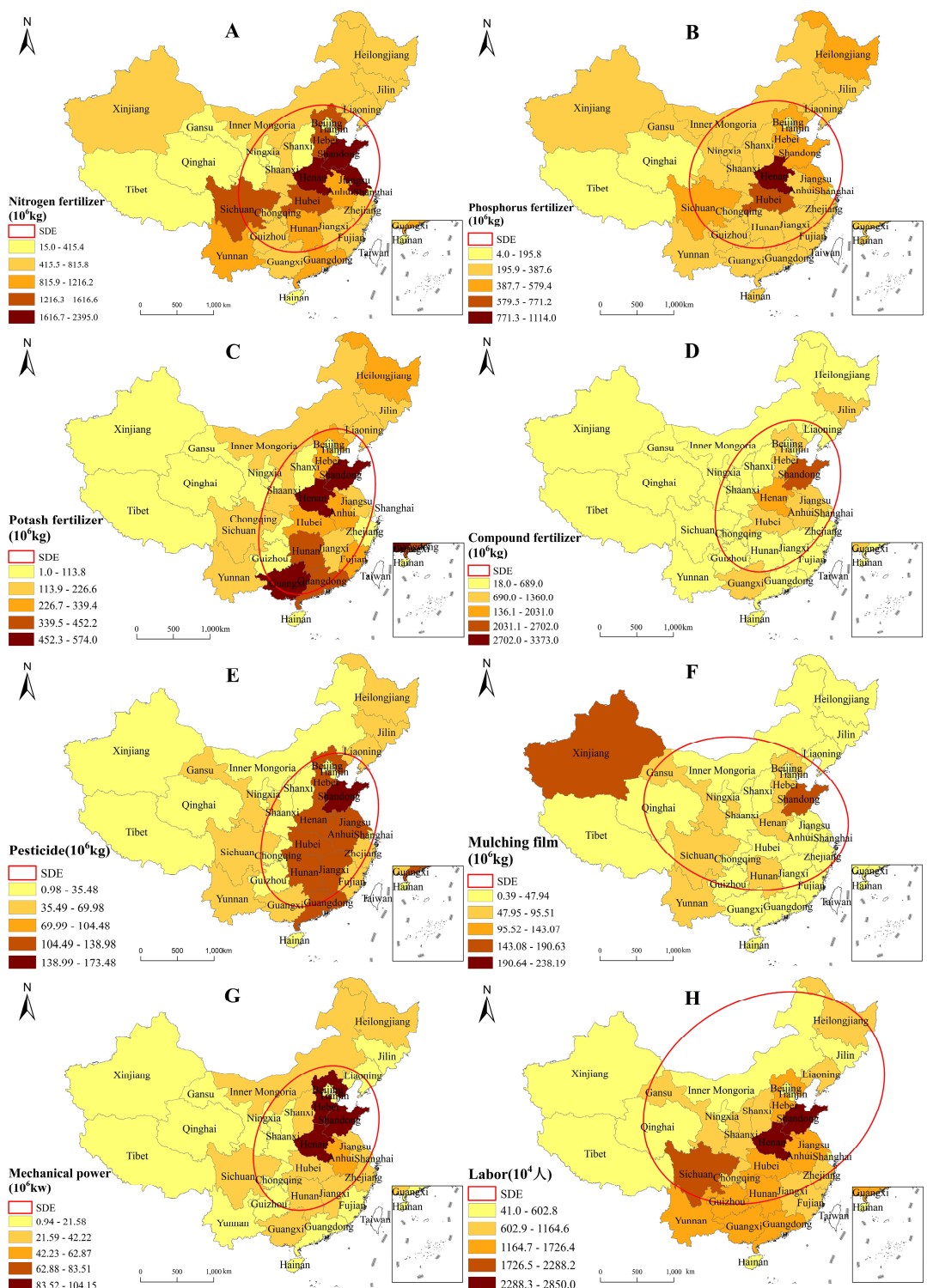

**Figure 2.** Input of production material on arable land in 2008.

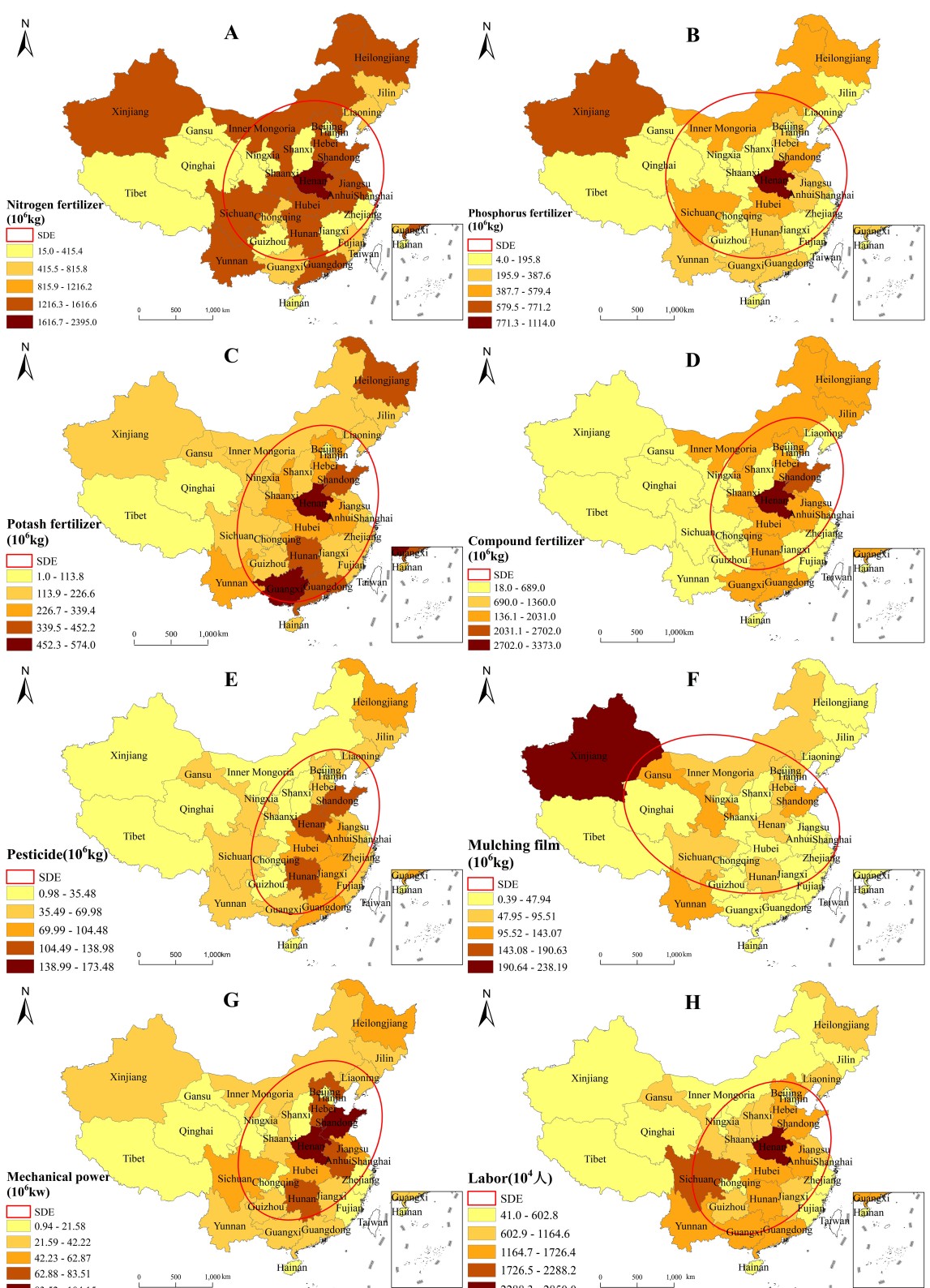

**Figure 3.** Input of production material on arable land in 2018.

In terms of specific inputs, in contrast to the distribution of total input (Figures 2 and 3), the inputs of nitrogen fertilizer and pesticide per unit area are increasingly concentrated in five provinces along the southeast coast (including Jiangsu, Zhejiang, Shanghai, Min, and Guangdong). In addition, both phosphate fertilizer and the plastic film showed an

SDE with an obvious northwest orientation. Represented by the high value of phosphate fertilizer and plastic film in Xinjiang Province, the trend was further increased.

### 3.2. The Spatiotemporal Change in the Output on Arable Land

#### 3.2.1. Provincial Output of Crop Yield

In 2008–2018, the yield of grain, vegetable, fruit, and oil crops increased, in which the most obvious crops were grain and vegetable. As pillar crops, the grain yield and vegetable yield increased from $5.29 \times 10^{11}$ and $5.92 \times 10^{11}$ kg to $6.58 \times 10^{11}$ and $7.03 \times 10^{11}$ kg, respectively (the annual growth rates were as high as 2.44% and 1.87%, respectively). Conversely, the total output of cotton, sugar crops, and tobacco showed a downward trend, of which sugar crops decreased the most (from $1.34 \times 10^{11}$ to $1.19 \times 10^{11}$ kg, with an average annual decline rate of 1.11%). Tobacco showed the fastest decline rate, with the yield decreasing by $5.97 \times 10^8$ kg, with an average annual decline rate as high as 2.10%.

Figures 6 and 7 represent the output on arable land in 2008 and 2018, respectively. The subfigures A-G illustrate the output of grain, vegetable, cotton, oil crop, sugar crop, fruit and tobacco, respectively. From the perspective of the distribution pattern, each crop has obvious clustering directivity. Except for tobacco, sugar, and cotton, the yields of other crops on the Huang-Huai-Hai Plain accounted for more than 35%. Compared with other areas, the Huang-Huai-Hai Plain maintained a very high level of output. Specifically, the proportion of fruit yield in this region to the whole country was up to 50.44%~52.59% (Figure 6F). Among them, the output of grain crops (Figure 6A) in the main producing areas was as high as $3.99 \times 10^{11}$ kg, accounting for 75.50% of the national total, which increased to 78.74% in 2018, occupying an absolute advantage.

In addition, the high-value area of grain yield shifted to the north. With Qinling Mountain and the Huaihe River as the dividing line, the ratio of grain yield between North China and South China was 54:46 in 2008, which changed to 59:41 in 2018, forming a pattern of high north–low south grain production. The SDE of tobacco (Figures 6G and 7G) showed an obvious southwest directivity. The output of tobacco in Yunnan and Guizhou provinces accounted for 44.44% of the total tobacco output in the country, which increased to 48.91% in 2018. In contrast, the SDE of cotton (Figure 6C) shows northwest directivity. In 2008, there were two cotton agglomeration areas in Xinjiang and Huang-Huai-Hai Plain, with the output accounting for 40.39% and 42.75% of the country, respectively. By 2018, the proportion of the cotton output of Xinjiang Province to the country increased to 83.75%, indicating an increasing spatial polarization phenomenon.

#### 3.2.2. Output of Crop Yield Per Unit Area

The crop yield per unit area of most crops (except sugar crops) showed the same changing trend as that of total crop output. Four crops continued to increase in both total yield and yield per unit area, including grain, vegetable, fruit and oil crops. Although the total amount of grain crops increased the most, the yield per unit area of grain did not increase significantly (from 4990.59 to 5580.04 kg/hm$^2$ in 2008–2018, with a growth rate of 11.81%). Oil yield per unit area increased the least (only increased by 385.61 kg/hm$^2$), yet it showed the highest growth rate (17.62%). It is worth noting that although the total output of sugar decreased, the yield per unit area of sugar increased significantly (from 40,646.39 to 43,487.71 kg/hm$^2$ in 2008–2018), showing the highest productivity per unit area. The changes in other crops were smaller (growth rate <10%).

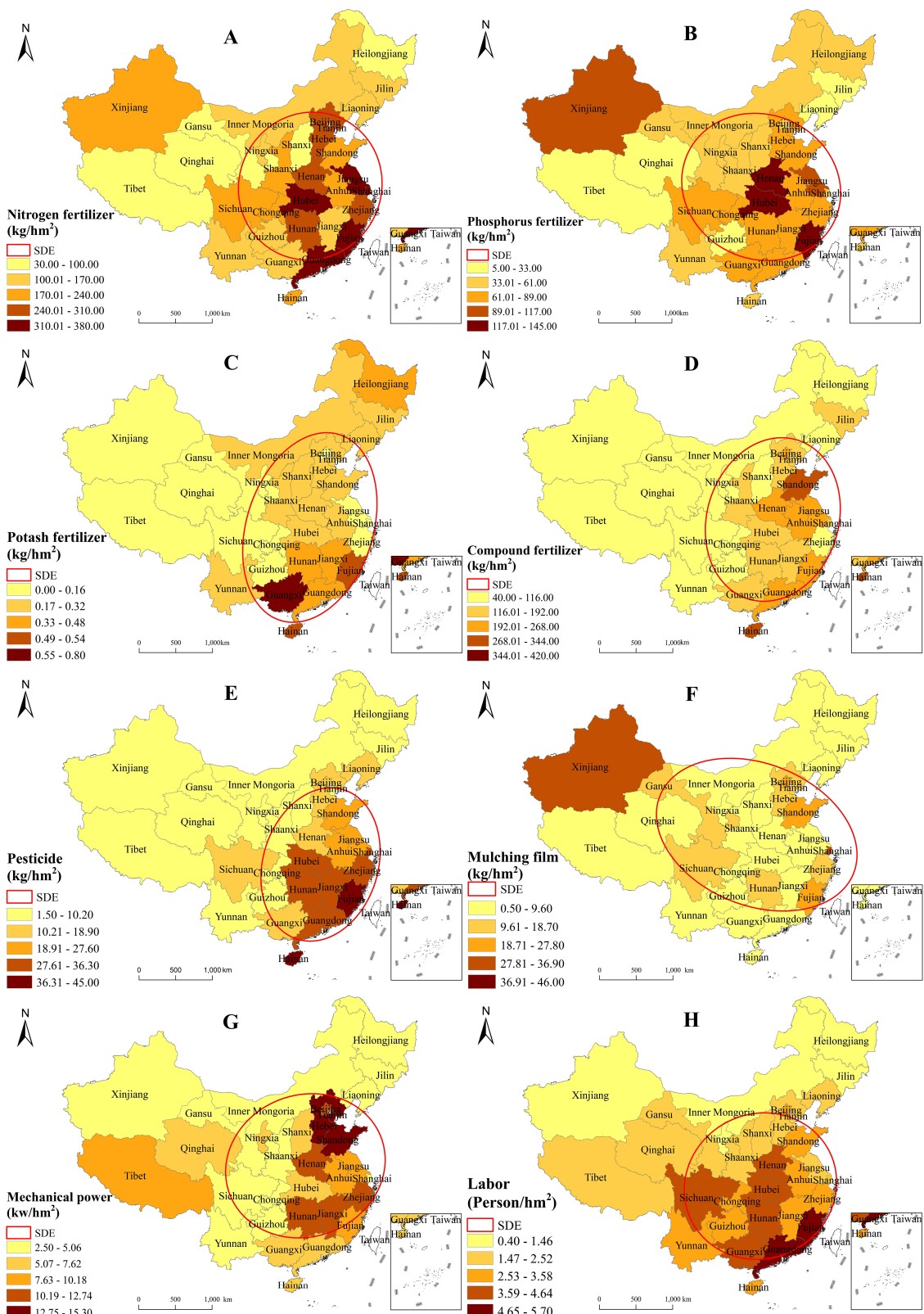

**Figure 4.** Input of production material on per unit area of arable land in 2008.

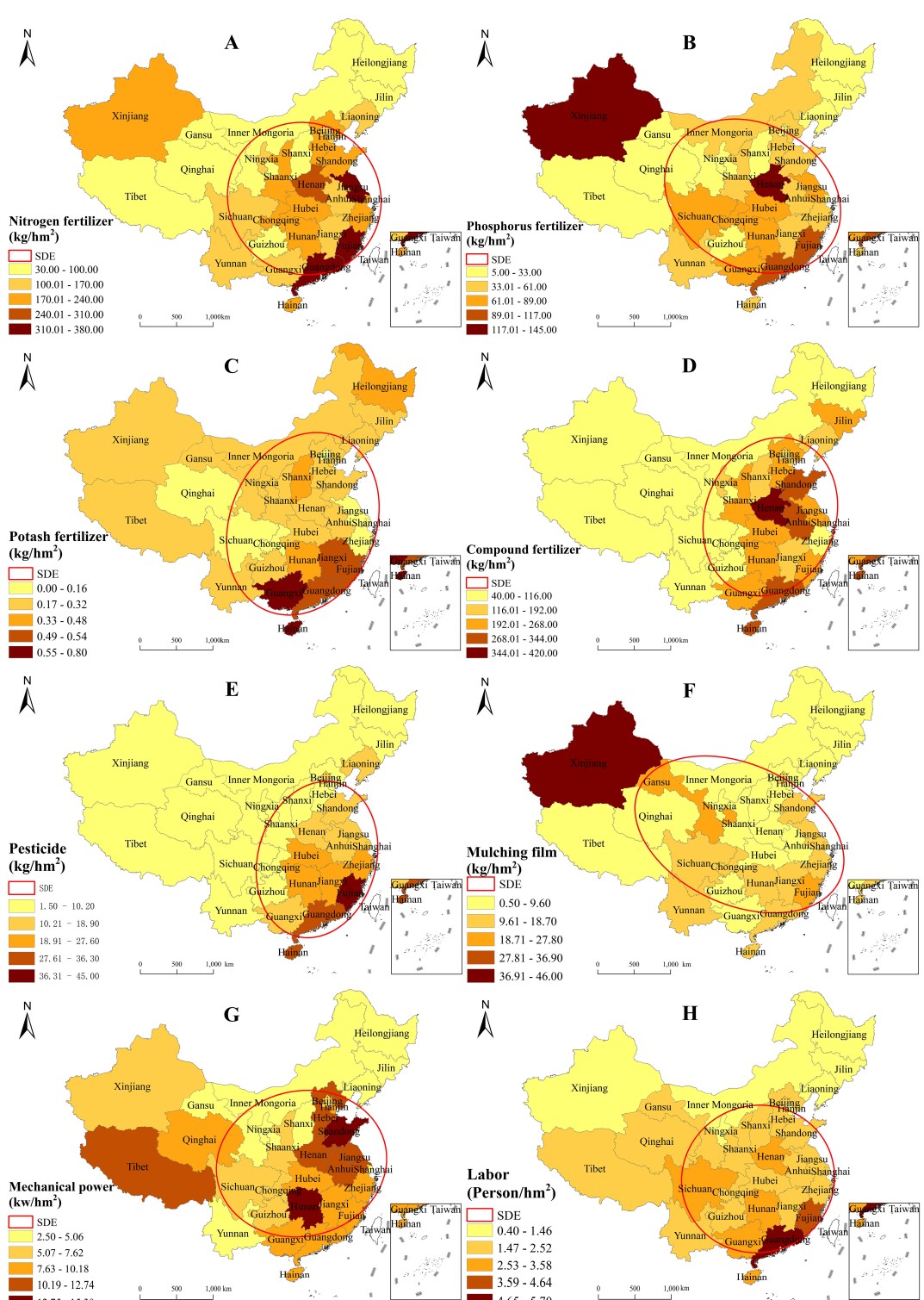

**Figure 5.** Input of production materials on per unit area of arable land in 2018.

Figures 8 and 9 represent the output on per unit area of arable land in 2008 and 2018, respectively. The subfigures A-G illustrate the output of grain, vegetable, cotton, oil crop, sugar crop, fruit and tobacco on per unit area, respectively. Interestingly, the spatial distribution of output per unit area and total output in each province are inconsistent. Taking grain as an example, although the grain yield of most major producing areas was higher than that of nonmajor grain producing areas (the former was 5464.21 kg/hm$^2$, while the latter was 4720.75 kg/hm$^2$), there are contrasting examples. The total grain

yield of Heilongjiang in the main grain-producing areas ranked 1st in the country, but its grain yield per unit was low (rank 24th in 2018). Conversely, the total grain outputs of Tibet, Guangdong, Fujian, and Zhejiang (members of nonmajor grain-producing areas) are low, yet their grain yields per unit area are relatively high. The distribution patterns of vegetables and fruits were similar: the high-value regions were concentrated in northern China. Even so, the yield per unit area of fruit still gradually formed a new high-value region in northeast China by 2018.

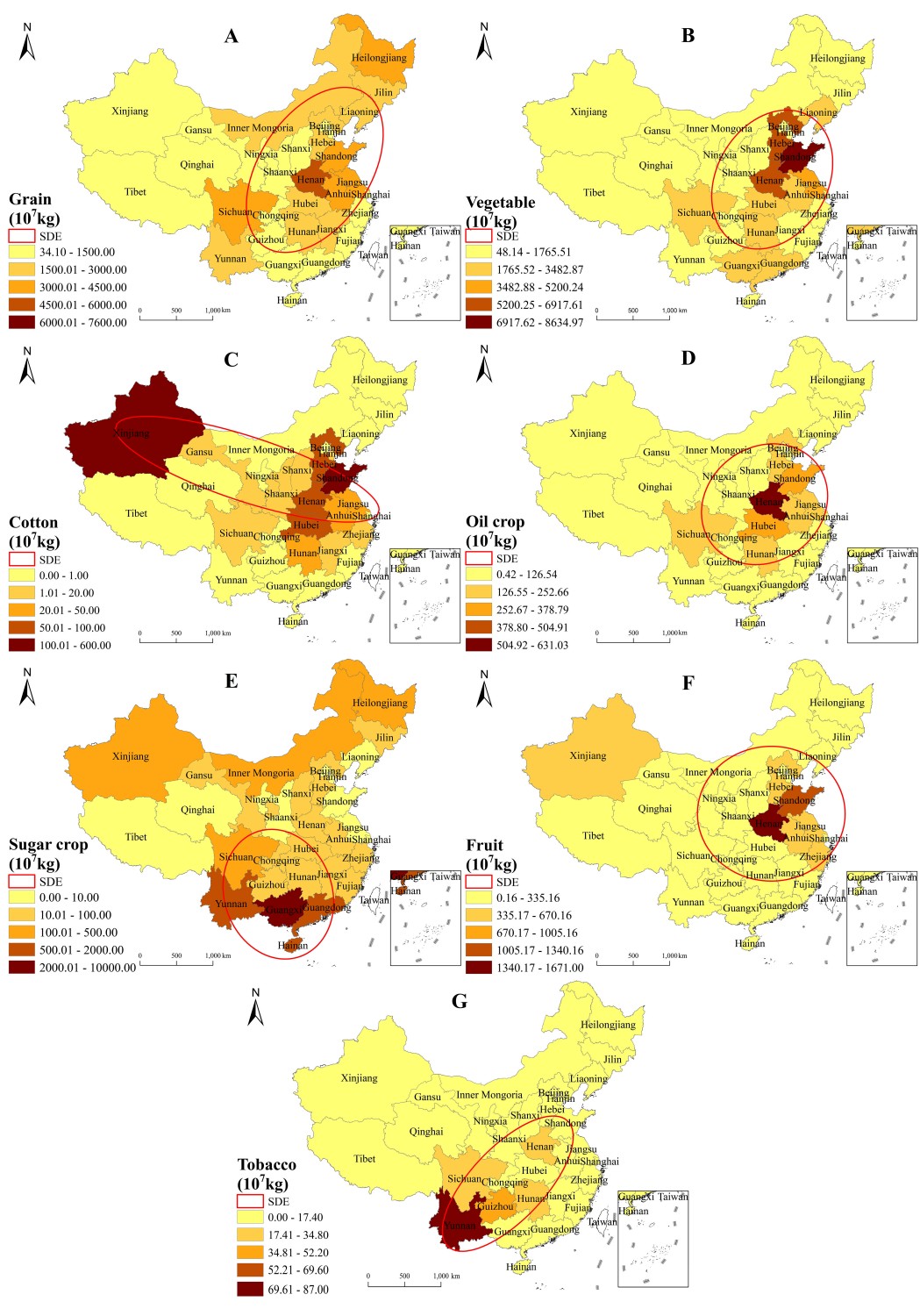

**Figure 6.** Output of crop yields on arable land in 2008.

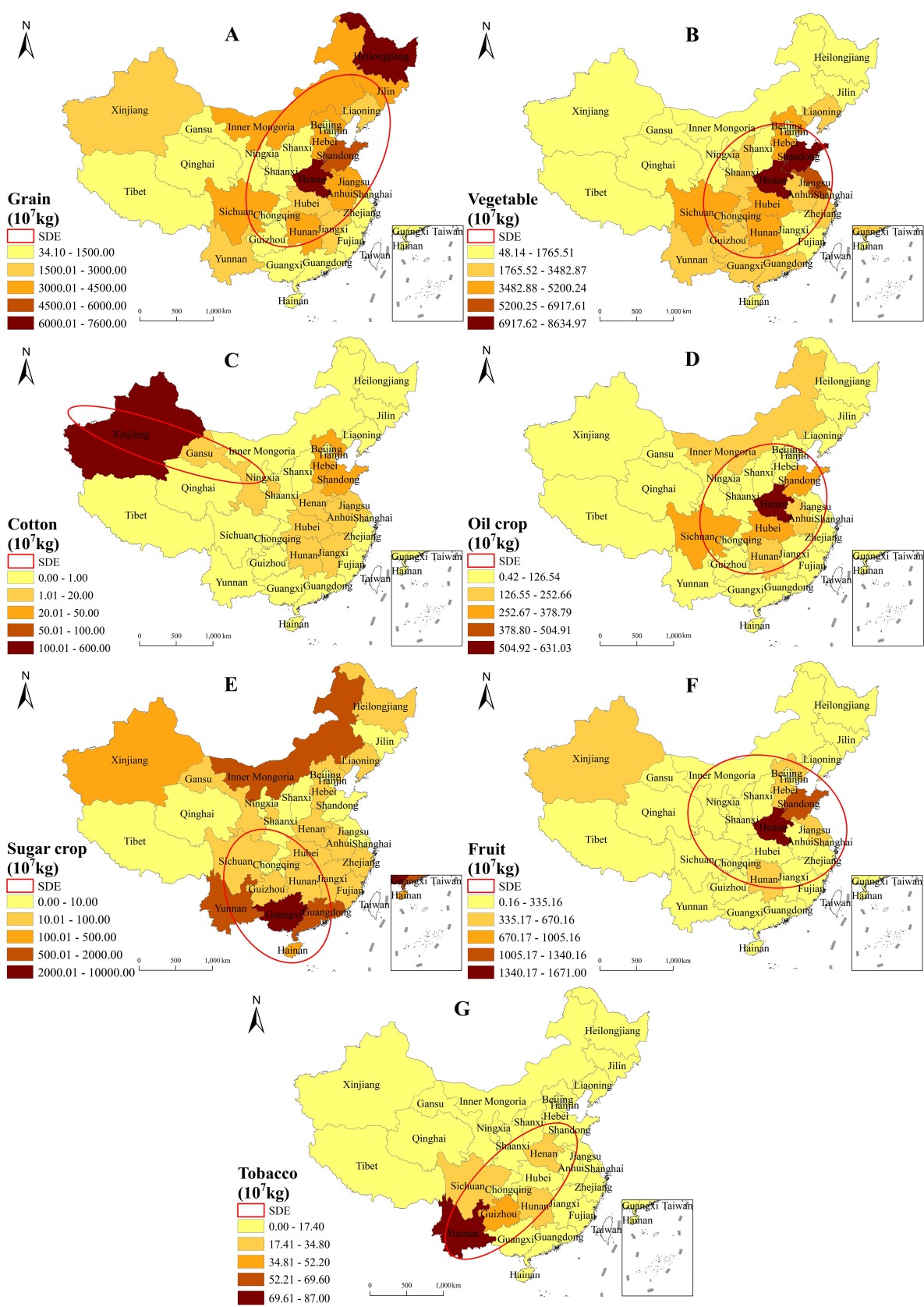

**Figure 7.** Output of crop yields on arable land in 2018.

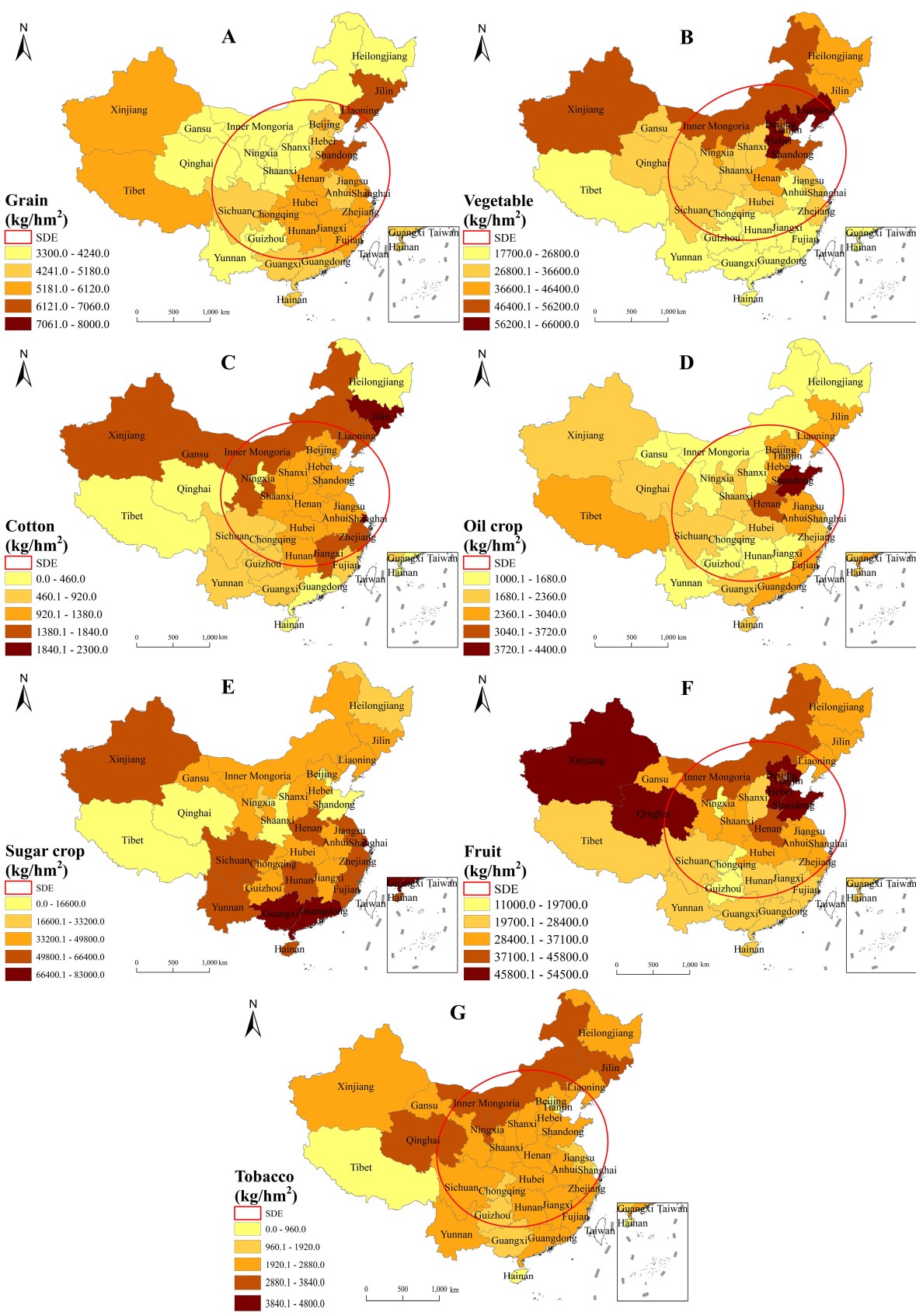

**Figure 8.** Output on per unit area of arable land in 2008.

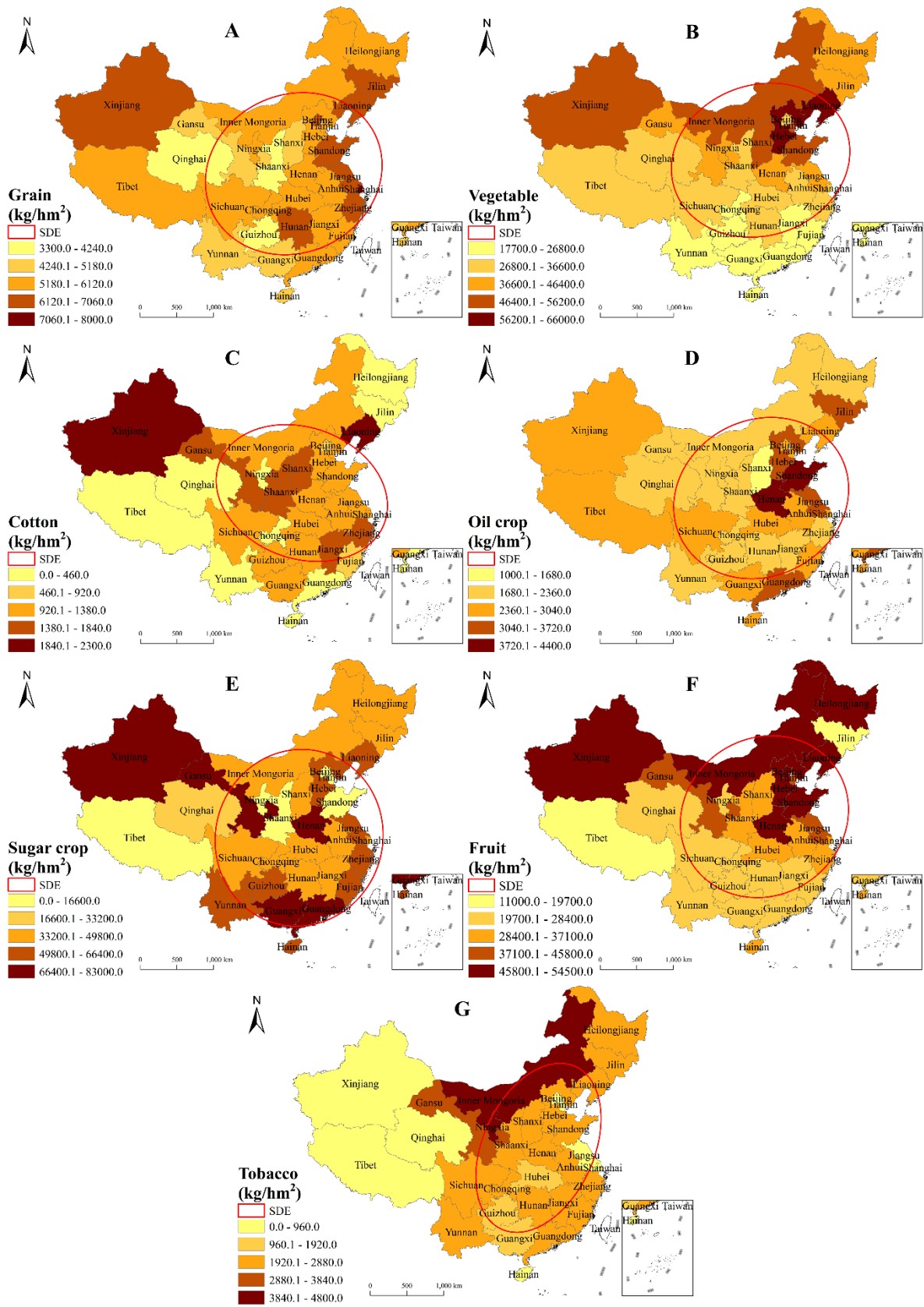

**Figure 9.** Output on per unit area of arable land in 2018.

### 3.3. The Movement of Spatial Barycenter for the Input and Output on Arable Land

The movement of the spatial barycenter for the input on arable land is shown in Figure 10, in which subfigure a and c illustrate the movement of barycenters of different kinds of inputs, while b and d illustrate the movement of barycenters of different inputs on per unit area. The spatial barycenters of all kinds of inputs were located in the central region of China (Henan, Hubei, and Shaanxi) and moved towards the west, which showed obvious consistency. In terms of the total amount of input (the red arrows in Figure 10), the spatial barycenter migrated towards the northwest. However, in terms of the input per unit area (the blue arrows in Figure 10), except for potash fertilizer and compound fertilizer, the spatial barycenter of all inputs moved towards the southwest.

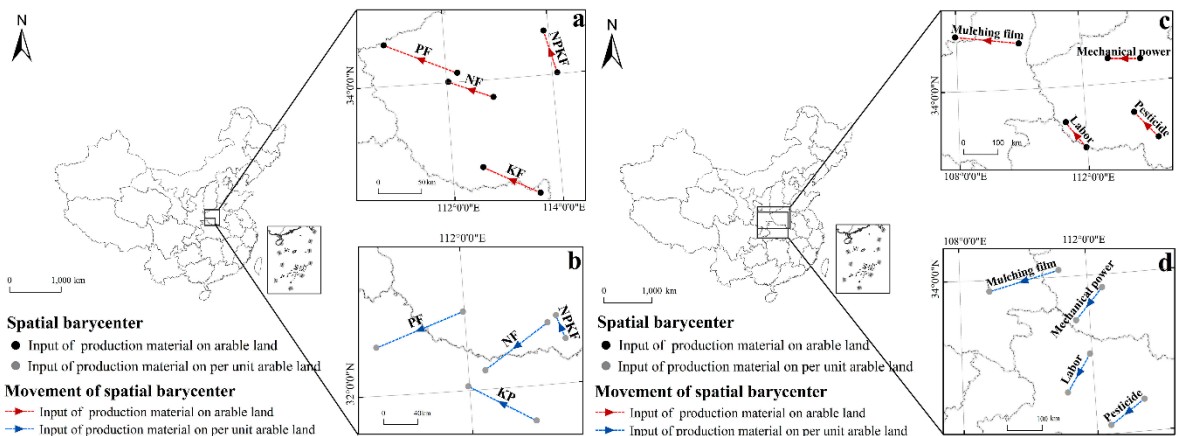

**Figure 10.** The movement of the spatial barycenter for the input on arable land (NF refers to nitrogen fertilizer; PF refers to phosphorus fertilizer; KF refers to potash fertilizer; NPKF refers to compound fertilizer).

In terms of movement distance, the spatial barycenter of nitrogen fertilizer moved 77.52 km westward and 25.63 km northward. The spatial barycenter of phosphate fertilizer moved 126.07 km westward and 46.99 km towards northward. The spatial barycenter of potash fertilizer moved 98.05 km westward and 43.54 km northward. Compound fertilizer moved 23.74 km westward and 71.62 km northward, respectively. Mulching film moved 176.95 km westward and 16.11 km northward. Pesticide moved 34.59 km westward and 39.08 km northward. Mechanical power moved 41.77 km westward and 1.09 km northward. Labor moved 25.66 km westward and 22.66 km northward. In total, the furthest movement westward is mulching film, the furthest movement northward is phosphorus fertilizer, and the furthest movement southward is mechanical power. Among all kinds of production inputs, the spatial barycenter of the plastic film showed the longest migration distance (177.68 km) and the fastest migration speed, and the linear migration distance was as high as 177.68 km. Pesticides showed the shortest movement distance (only 34.24 km) and the slowest moving speed.

The movement of the spatial barycenter for the outputs on arable land is shown in Figure 11, in which subfigure a and c illustrate the movement of barycenters of different kinds of outputs, while b and d illustrate the movement of barycenters of different outputs on per unit area. From the perspective of total output (the red arrow in Figure 11), the barycenter of grain moved towards the northeast, cotton and sugar crops moved towards the northwest, and the other crops all moved towards the southwest. In terms of yield per unit area (the blue arrow in Figure 11), only the fruits, vegetables, and sugar crops moved in the same direction as their total yield. The barycenter of both total output and yield per unit area of fruits and vegetables moved towards the southwest. Conversely, the barycenter of both total output and yield per unit area of sugar crops shifted towards the northwest.

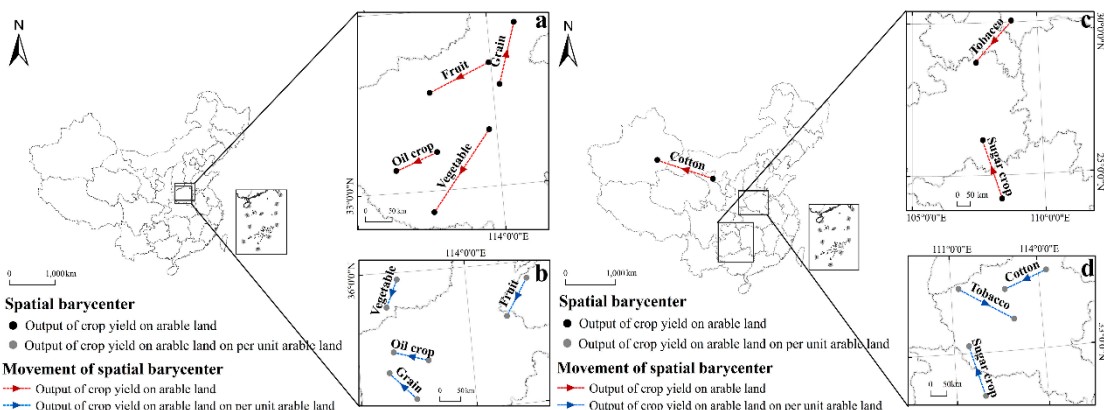

**Figure 11.** The movement of the spatial barycenter for the outputs on arable land.

In terms of movement distance, the spatial barycenter of grain moved 27.75 km eastward and 118.25 km northward. Vegetables migrated 103.49 km westward and 156.79 km southward. Fruits moved 111.48 km westward and 57.87 km southward, and oil crops moved 77.07 km westward and 35.96 km southward. Tobacco moved by 80.90 km westward and 101.64 km southward. Sugar crops moved 28.61 km westward and 37.24 km northward. Cotton moved 1112.35 km westward and 358.34 km northward. In total, in the east–west and north–south directions, the spatial barycenter of cotton moved by the longest distance (1168.64 km, with a speed of 116.8 km/year). Conversely, sugar crops moved by the shortest distance (46.96 km, with a speed of 4.70 km/year).

### 3.4. Degree of Coupling Coordination for the Input–Output on Arable Land

The degree of coupled coordination of input–output (D_CCIO) in 2008 and 2018 was illustrated in Figure 12a,b, respectively. The D_CCIO on arable land was low but increased gradually (average D_CCIO increased from 0.50 to 0.52 in 2008–2018). In terms of the grades of D_CCIO, the number of high-grade provinces increases while the number of low-grade provinces decreases.

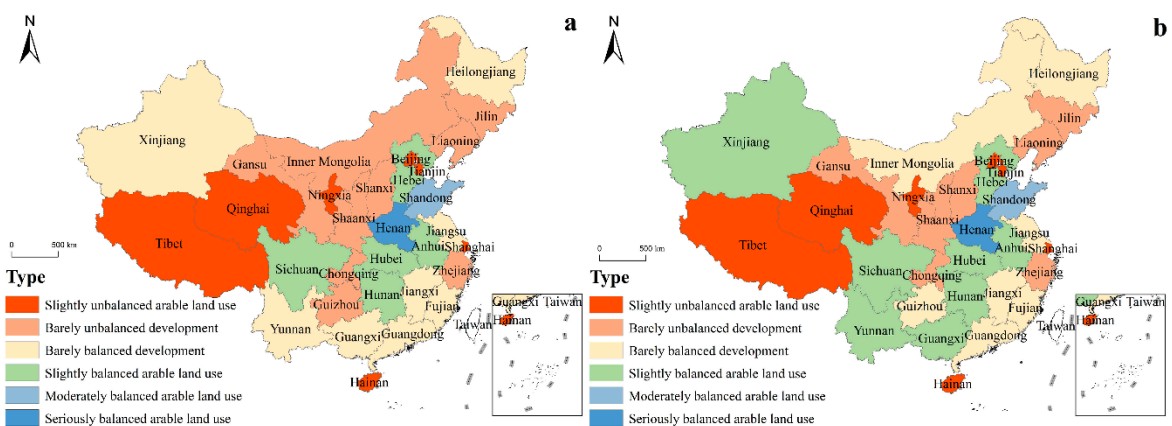

**Figure 12.** The change of D_CCIO among different provinces in 2008–2018.

Specifically, Henan and Shandong were the only provinces that showed seriously balanced and excited balanced grades. More provinces showed slightly balanced grades, including Hebei, Anhui, Hubei, Hunan, and Sichuan in 2008, with three newly added members (Guangxi, Yunnan, and Xinjiang) in 2018.

The number of provinces with barely balanced D_CCIO decreased. In 2008, Xinjiang, Guizhou, Yunnan, Guangxi, Heilongjiang, Jiangsu, Jiangxi, Fujian, Guangdong, and Guizhou were excluded by 2018. The number of provinces with barely unbalanced D_CCIO

decreased, including Inner Mongolia, Jilin, Liaoning, Gansu, Shaanxi, Shanxi, Chongqing, Guizhou, and Zhejiang, among which Inner Mongolia and Guizhou were excluded. The provinces with the lowest D_CCIO remained unchanged, including Tibet, Qinghai, Ningxia, Beijing, Tianjin, Shanghai, and Hainan. By comparing the input index and output index, except Yunnan, Guizhou, and Shanghai, the output of all provinces lags behind the input.

In terms of the change in D_CCIO (Figure 13a, a represents the change of input-output, and b represents the change of D_CCIO), only 5 provinces experienced changes in the D_CCIO grades, including Inner Mongolia, Xinjiang, Yunnan, Guizhou, and Guangxi. In detail, by comparing the change of input and output, this study found five types of changing patterns in different provinces (Figure 13b): (1) the output grew faster than the grow of input, (2) the input index grew faster than the grow of output, (3) the input declined faster than the decline of output, (4) the input decreased whereas the output increased, and (5) the input increased whereas the output decreased. The 1st and 4th changing patterns were concentrated in South China, whereas the 2nd changing pattern was concentrated in Northwest China, and the 3rd pattern was concentrated in the North China Plain.

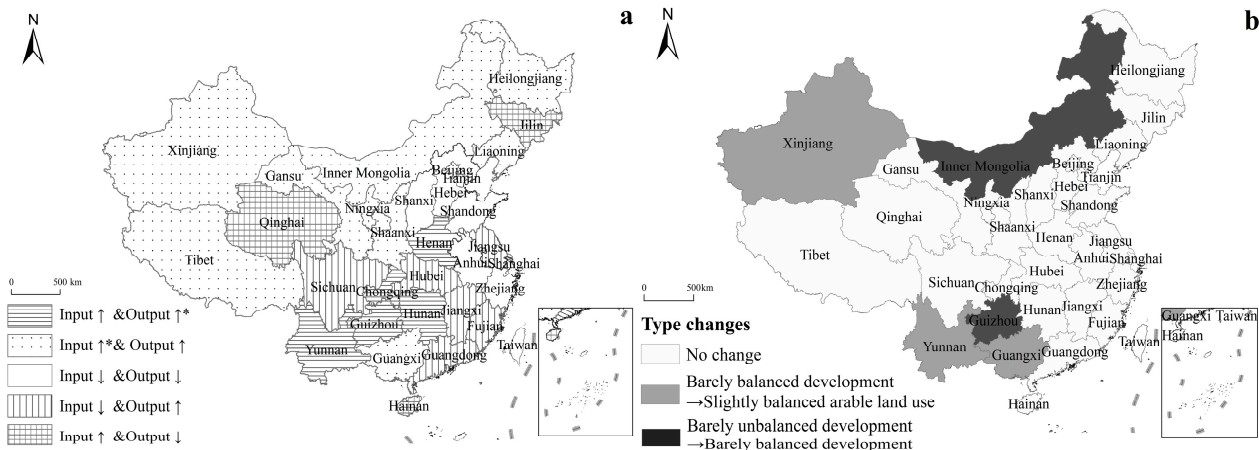

**Figure 13.** Change of the input–output and the coupling coordination type (the * in Figure 12a indicates the one which changed more. For instance, input ↑ and output ↑ * means the output increased more than the increase of input).

The research results found 13 "excellent provinces" that have good farmland utilization patterns (marked with * in Table 3). First, Henan and Shandong continued to show the highest D_CCIO from 2008–2018, which played a leading role in high-efficiency arable land use. Second, Yunnan, Guizhou, Guangxi, Xinjiang, and Inner Mongolia showed a significant increasing trend of D_CCIO, which indicated that the arable land use pattern was improving towards higher coordination among input and output. Third, Sichuan, Hubei, Jiangsu, Jiangxi, Fujian, and Southern Guangdong provinces showed an "input drops and output increases" trend, which proved a changing trend of arable land use towards higher productivity with lower input. These "excellent provinces" illustrated reasonable arable land use patterns towards better coordination between input and output, which was beneficial for the long-term use of scarce arable land.

**Table 3.** Different changing mode of degree of coupling coordination between the input and output (D_CCIO) among the 31 provinces.

| Level of D_CCIO | Change of Input (I) and Output (O) | Characteristics | Provinces |
|---|---|---|---|
| Seriously balanced | Both I and O increased | ΔI < ΔO | Henan * |
| Moderately balanced | Both I and O decreased | ΔI > ΔO | Shandong * |
| Slightly balanced | Both I and O increased | ΔI < ΔO<br>ΔI > ΔO | Yunnan *, Hunan<br>Xinjiang *, Guangxi * |
|  | Both I and O decreased | ΔI > ΔO<br>ΔI = ΔO | Hebei<br>Anhui |
|  | I and O changed reversely | I decreased and O increased | Sichuan *, Hubei * |
| Barely balanced | Both I and O increased | ΔI < ΔO | Guizhou * |
|  |  | ΔI > ΔO | Inner Mongolia *, Heilongjiang |
|  | I and O changed reversely | I decreased and O increased | Jiangsu *, Jiangxi, Fujian, Guangdong |
| Barely unbalanced | Both I and O increased | ΔI < ΔO<br>ΔI > Δ O | Chongqing<br>Gansu, Shaanxi, Liaoning × |
|  | Both I and O decreased | ΔI > ΔO | Shanxi, Zhejiang |
|  | I and O changed reversely | I increased and O decreased | Jilin × |
| Slightly unbalanced | Both I and O increased | ΔI > ΔO | Tibet, Ningxia |
|  | Both I and O decreased | ΔI > ΔO | Beijing, Tianjin, Shanghai |
|  | I and O changed reversely | I increased and O decreased | Qinghai, Hainan |

Note: * refers to the increase in the efficiency of the use of arable land and × refers to input increased and output decreased.

However, some of these "excellent provinces" have objective limits on arable land use. For instance, Yunnan, Guizhou, and Fujian are restricted by local mountainous terrain. Although plenty of rain and light exists in Sichuan, mountainous areas concentrated in western Sichuan bring a mountainous plateau climate, which causes insufficient heat for arable land use (annual temperature is only 4~12 °C). Moreover, as one of the most economically developed areas in China, Guangdong shares a low agricultural contribution rate (agricultural output accounted for only 3.94% in the province by 2018). Except for Henan, Shandong, Inner Mongolia, and Xinjiang, these "excellent provinces" are distributed in southern China. This forms a similar distribution with the current economic pattern (higher nonagricultural economic output in southern China and lower in northern China), which may bring increasingly higher pressure for agricultural development in these areas. In the future, the rapid nonagricultural economic development in the "excellent provinces" may affect the arable land amount and quality and further disturb the input–output coordination on arable land.

In those provinces with the lowest D_CCIO, the lack of arable land resources is the main limitation, as megacities in China, Shanghai, Beijing and Tianjin had low arable land areas (lower than $4.37 \times 10^5$ hm$^2$), which ranked last, second last and third last among the studied provinces (Table 3). Located in the western high-altitude zone, Tibet and Qinghai lack ideal water and heat resources and maintain low cultivated land area ($<5.91 \times 10^5$ hm$^2$), which ranks 4th and 5th from the bottom, respectively. Hainan Province is also limited by the amount of arable land resources ($<7.22 \times 10^5$ hm$^2$), which ranked 6th from the bottom. Located on the Loess Plateau, Ningxia faces the challenge of the natural environment caused by water resource shortages and soil erosion (cultivated land area $<12.9 \times 10^5$ hm$^2$, the 7th from the bottom). These regions all showed a characteristic of a large change in input and a small change in output. Surprisingly, this study found that two major national agricultural regions showed significantly weaker land use:

Jilin and Liaoning Provinces (marked with $^\times$ in Table 3). As an important agricultural region in northeastern China, Jilin and Liaoning enjoyed a deal area of arable land (the former $4.972 \times 10^6$ hm$^2$, ranked 5th, the latter $6.987 \times 10^6$ hm$^2$, ranked 13th), whereas they showed lower D_CCIO and a less ideal change trend (input increase but output decrease in Jilin, output growth slower than the growth of input in Liaoning). The "output lags behind input" phenomenon in important agricultural production provinces may go against the overall arable land use efficiency.

## 4. Discussion

### 4.1. Differences in the Recessive Transfortion of Arable Land Use Patterns

Facing the dynamic changes of society–economy–ecology in rapidly urbanizing areas, land use transition often becomes a common practice. Previous studies have pointed out two mechanisms of land use transition: explicit transition and recessive transition [42]. The explicit land use transition is mainly manifested as regional construction land expansion and arable land contraction in rapid urbanization process [43,44]. The recessive transition of arable land use may be more invisible and complex [45], such as the recessive transition of the input structure on arable land [46]. By comparison, this study confirms the existence of recessive transition of arable land use from the perspective of input–output change. Because the 31 provinces belong to different physical geographic regions of China (Figure 1), the details of recessive transition of arable land use in this study can be discussed as follows.

First, located in North China District, the Hebei, Shandong, Henan showed a "high input and high output" characteristic, accompanied by internal structural transition of input. Most inputs in this region showed a downward trend (except compound fertilizer and mechanical power). Especially, the decrease in nitrogen fertilizer, pesticide, and labor input was more than 1.25 times the national average level, in which the decline rate of nitrogen fertilizer was close to twice the national average. However, the increase in mechanical power was only 20.56% of the national average. Although these areas enjoy unique natural advantage of flat terrain and rich farmland resources, affected by the traditional small farmer production mode, arable land patches are mostly fragmented and large-scale mechanized farming is inadequate [47,48]. In addition, the groundwater funnel caused by excessive groundwater exploitation for irrigation will also force these areas to seek water-saving arable land use patterns [49]. Therefore, the transition mechanism of arable land use in this area can be summarized as "increasing compound fertilizer and machinery power input, and decreasing the overall input, accompanied by incomplete mechanization".

In contrast, though Jilin, Liaoning, and Heilongjiang Province are located in North-East District, one of the most important agricultural production areas in China, showed low D_CCIO. Enjoying the fertile black soil and large-scale management of arable land, the grain sown area in these areas have been proved to increase continuously in recent years [50]. However, the growth rate of input was high (except labor force, all input factors increased by more than 20%, far exceeding the national average level), with a lag in the output on arable land. One of the reasons for this low D_CCIO in this area is the lag of yield per unit area. Specifically, the high total output but low output per unit area in this area have highlighted the lag of productivity per unit area, which is also a manifestation of the recessive transition of arable land use. Therefore, the recessive transition mechanism of arable land use in northeast China can be summarized as "increasing overall inputs and sparing labor force on the premise of expanding sown area, accompanied by lagging productivity per unit area".

Located in Central China, Hunan, Hubei, Anhui, Jiangxi, Jiangsu, Zhejiang and Fujian Province enjoy ideal climate condition (average annual precipitation is 1300–1900 mm, average annual temperature is 16~19 °C), meanwhile, fertile irrigation from the Yangtze River gives this area natural advantages of planting. However, the D_CCIO of this area is not the highest. The main reason is the difference of terrain and economic development mode. Specifically, abundant arable land on the plain in Hunan and Hubei give them the

chance to increase the ago-mechanical input significantly (with a growth rate of 58%). With mountainous and hilly areas accounting for 80%, Fujian and Jiangxi lack the advantage of expanding to the farming scale, which makes an overall decline in input and output growth. Therefore, the recessive transition mechanism of arable land use in this area can be summarized as "polarization of mechanization level and planting scale caused by terrain difference".

Though located in the North-West District with severe water shortage and diurnal temperature variation, Xinjiang showed a high D_CCIO in a typical oasis agricultural mode. A closer look reveals that the total inputs in this region are not high, but the use of mulching film and phosphate fertilizer is very prominent, ranking 1st and 2nd among the 31 provinces, respectively. The main reason comes from the predominant crop (the total yield, per unit yield and sown area of cotton in Xinjiang ranks 1st in China). As two highly needed factors during the growth of cotton [51,52], mulching film and phosphorus fertilizer become the most significant input on arable land. Similarly, located in South China District, Yunnan enjoys subtropical monsoon climate with sufficient water-heat resources and fertile red soil, which makes ideal living environment for tobacco and sugar crops [53]. Potash fertilizer, the indispensable factor for tobacco and sugar crops, showed a growth rate with 7.5 times that of the average level in China. Superior crop in Xinjiang and Yunnan shows a similar influence on the input–output pattern on arable land. As a result, the transition mechanism of arable land use in these two regions can be summarized as "specializing plantation of predominant crop, with priority of resource input and production output."

Located in the Qinghai-Tibet District with an average altitude above 4000 m, Qinghai and Tibet have a typical plateau mountain climate. Water, heat and arable land are scarce in this area, which limited the arable land use and make the D_CCIO extremely low. Though far lower than the national level, most inputs kept increasing. However, natural constraints hinder the increase of output on arable land. Under such objective constraints, the transition mechanism of arable land use in this region can be summarized as "retreat from agricultural development and weaken the use of arable land".

To sum up, significant different transition mechanisms of arable land use among different provinces are hidden in the internal structural change of input–output on arable land [20]. The constraints of arable land use lay in the deficient agricultural mechanization input caused by the smallholder management mode, the lagging productivity per unit area, and the polarized input–output pattern caused by dominant crop planting. To improve the efficiency and rationality of arable land use, the basic premise of "taking measures according to local conditions" is needed. Characteristics of regional geographical conditions and planting structures should be taken into deeper consideration, which will be a necessary mean for optimized transition of arable land use.

### 4.2. Plenty of Room for Optimizing the Planting System on Arable Land

Based on microscale experimental research, previous studies have emphasized the importance of optimizing planting systems, which may contribute to the efficiency of comprehensive resource use [54,55]. In contrast, based on a geographical spatiotemporal perspective, this study proves that there is plenty of room for optimizing the planting system. The spatial distribution of total output and output per unit area differs greatly, which may be a breakthrough point for the optimization of planting structure. In addition, the movement of spatial barycenters of the total output and output per unit area did not match well, which also confirms the necessity of planting structure optimization. The movement of the spatial barycenter of the total grain yield towards the northeast and the grain yield per unit area towards the northwest are the east-west opposite. The movement of the spatial barycenter of the total oil crop yield towards the southwest and its yield per unit area towards the northwest are south–north opposite. The total yield of tobacco towards the southwest and the yield per unit towards the southeast are the east–west

opposite. Similarly, the total yield of cotton towards the northwest and the yield per unit area towards the southwest are south-north opposite.

Theoretically, the yield per unit area could better reflect the productivity of arable land. The "opposite movement" of the total yield and yield per unit area is probably a result of the unreasonable allocation of cultivated land scale and planting structure. Taking Heilongjiang and Xinjiang as examples, Heilongjiang's total grain output ranks 1st in China, while its per unit area yield ranks 24th. The total grain output of Xinjiang ranks 15th in China, while its per unit area yield ranks 2nd. This large contrast may be caused by regional resource endowment differences and agricultural scales. Although the base of arable land area in Xinjiang is small (5148.1 $hm^2$), the inland rivers supplied by melting water from snow in this area provide up to 94.86% irrigation. Although Heilongjiang has as much as 15,845.9 $hm^2$ of arable land, its effective irrigation rate is only 38.62%. In addition, the long sunshine time, high photosynthetic efficiency, and sparse population make Xinjiang a flexible place for arable land use. "Plant crops if the land is good, and graze if the land is bad," which makes the average yield per unit area in Xinjiang enjoy certain advantages.

The mismatch between the total yield and yield per unit indicated that the increase in crop yield might rely heavily on the increase in sown area rather than land productivity, which emphasizes the importance of optimizing the planting structure. Previous research also shows that there is no significant scale benefit in arable land, i.e., expansion of arable land or sown area does not necessarily lead to an increase in crop yield [56]. From this perspective, this study found break-through points for the places where total yield and per unit yield were uncoupled. The grain of Xinjiang, the vegetables of Inner Mongolia, Xinjiang, Liaoning and Hebei, the cotton of Jilin, the oil crops of Hebei, Guangdong and Jilin, the fruits of Inner Mongolia, Heilongjiang, Liaoning and Hebei, and the tobacco of Inner Mongolia and Gansu all show typical uncoupling characteristics of "high yield per unit area but low total yield." In response, based on local arable land resources, gradually transforming the planting structure to these special local crops may become an important means for the transition and optimization of arable land use.

*4.3. Policy Implications of Arable Land Use Transition*

To better improve national arable land use on a macroscale, the results pointed to some possible policy-making points. First, in view of the recessive transition of the input–output structure of arable land use, optimizing the mode of arable land utilization by taking measures according to local conditions should be taken seriously. Henan and Shandong should strengthen the protection of arable land amount and give full play to agro-mechanization. For those with low input and low output levels, such as Tibet, Ningxia, Beijing, Tianjin, Shanghai, Qinghai, and Hainan, which have disadvantages in arable land scale, they should appropriately reduce the intensity of agricultural production and take regional economic transition as the main direction. For those with high input-low output levels, such as Inner Mongolia, Heilongjiang, Gansu, Shaanxi, and Liaoning, they should focus on enriching the soil fertility, improving the irrigation rate, and reasonable fallowing.

Second, long-term arable land use planning should focus on long-term observations of productivity among different regions. The grain of Xinjiang, the vegetables in Inner Mongolia, Xinjiang, Liaoning and Hebei, the cotton in Jilin, etc., all show a decoupling phenomenon of high yield per unit and low total yield, which highlights the necessity of long-term monitoring on a macro scale. Dynamically allocating agricultural production tasks on the basis of long-term observations can help to maximize the productivity of arable land and give full play to regional advantages.

Finally, the policy emphasis on the transition of arable land use lies in the coordinated relationship between humans and land. China's rapid urbanization caused more than 200 million farmers to leave arable land and seek jobs in cities in 1978–2018, which may further affect national arable land use efficiency [2–4,57]. Scholars insist that arable land use should focus on saving arable land and shifting from labor-intensive land to technology-

intensive land [18]. However, in areas with relatively abundant arable land resources, such as Northeast China and Xinjiang, there is still much potential for the large-scale management of arable land. In these areas, either bringing labor back to arable land or letting in capital and markets, the formation of specialized farm operations, and widely improving the technical advantages of the use of arable land are suitable paths for the region. In contrast, in the North China Plain with many small farmers, the policy should focus on the promotion of arable land circulation. It should provide professional farmers with flexibility to expand the arable land area and give stable contract rights to those farmers who rent out their arable land. On the other hand, in the hilly mountainous areas of Southwest China, policy-making should focus on preventing arable land abandonment, prohibiting the occupation of flat farmland, and supplementing sloping arable land to increase farmers' enthusiasm for arable land use [58].

## 5. Conclusions

By analyzing the degree of coupling coordination (D_CCIO) of input–output on arable land, this study pointed out different recessive transitions of arable land use among 31 provinces in mainland China. First, the total input of arable land and the input per unit area showed different spatial-temporal changes. Although the total input and input per unit area of nitrogen fertilizer, pesticide, and labor force all decreased, the total input and input per unit area of compound fertilizer, mulching film, potash fertilizer, and mechanical power increased. The total input of phosphorus fertilizer increased, while its input per unit area decreased. The spatial barycenter of the total inputs moved towards the northwest, while the input per unit area moved in another direction (southwest, except potash and compound fertilizers).

Second, the total output and the output per unit area also showed spatial-temporal disparities in arable land. The total output and yield per unit area of grain, vegetables, fruit, and oil crops increased, yet the total yield and yield per unit area of tobacco and cotton decreased. The total yield of sugar crops decreased, but the yield per unit area increased. The spatial barycenter of total grain output moved towards the northeast, but the grain yield per unit area moved to the northwest. The spatial barycenter of the total output of oil crops moved towards the southwest, per unit area of yield to the northwest. The total yield of tobacco moved towards the southwest, while the yield per unit area moved towards the southeast. The total cotton output moved towards the northwest, while the output per unit area moved towards the southwest.

Third, the recessive transition mechanism of arable land use was hidden in the internal change of the input–output structure. Although the D_CCIO increased overall, the output of most provinces lagged behind the input. Specifically, Henan and Shandong showed the highest D_CCIO, with a transition mechanism of "increasing compound fertilizer and machinery power input, and decreasing the overall input, accompanied by incomplete mechanization." Northeast China (Liaoning, Jilin, Heilongjiang) showed lower D_CCIO, with a recessive transition mechanism of arable land use of "increasing overall inputs and sparing labor force on the premise of expanding sown area, with lagging productivity on per unit area." Xinjiang in northwest China showed a transition mechanism of arable land use in this area characterized by "specializing plantation of predominant crop, with priority of resource input and production output."

Finally, this paper proposes a policy proposal for optimized arable land use patterns. On the one hand, the lagging of per unit area yield in Northeast China, the incomplete mechanization level on the Huang-Huai-Hai Plain, and the support of dominant characteristic crop planting in Xinjiang should be improved. On the other hand, considering the spatial decoupling total yield and yield per unit area, the planting structure should be adjusted according to local advantages. The grain of Xinjiang, the vegetables of Inner Mongolia, Xinjiang, Liaoning and Hebei, the cotton of Jilin, the oil crops of Hebei, Guangdong and Jilin, the fruits of Inner Mongolia, Heilongjiang, Liaoning and Hebei, and the tobacco of Inner Mongolia and Gansu with "high yield per unit and low total yield" should be given

priority. In addition, in view of the human-land relationship, specialized large-scale arable land management should improve in regions with rich arable land resources (northeast China and Xinjiang agricultural reclamation corps). In the fragmented arable land on the North China Plain, stable contract rights should be improved. Smallholders should be encouraged to freely transfer their arable land to professional farmers with large-scale land patches to improve mechanization in this area. In the hilly and mountainous areas of Southwest China, it is necessary to avoid compulsory large-scale land consolidation to avoid the loss of flat arable land and land abandoned by farmers. Based on the above research conclusions, this paper provides a detailed scientific reference for the observation of arable land transition and land use optimization in China.

Limited as it is by provincial data, this study cannot explore the spatial-temporal pattern of recessive transition of arable land use in finer scale. In the future, deeper studies based on city-level or county level are highly recommended to reflect the possible different recessive transitions mechanisms of arable land use in China or other countries.

**Author Contributions:** Conceptualization, Y.L. and G.Y.; methodology, Y.L. and G.Y.; software, Y.L., G.Y. and Y.X.; validation, Y.L., G.Y. and Y.X.; formal Analysis, G.Y.; investigation, Y.L., S.L. and G.L.; resources, G.Y. and X.W.; data Curation, Y.L.; writing—original draft preparation, Y.L.; Y.X. and S.X.; writing—review and editing, Y.L. and G.Y.; visualization, Y.L. and Y.X.; supervision, G.Y.; project administration, G.Y. and Y.L.; funding acquisition, G.Y. All authors have read and agreed to the published version of the manuscript.

**Funding:** This paper was funded by the National Natural Science Foundation of China (Project No. 41701590); Humanities and Social Sciences Foundation of the Ministry of education, China (Project No. 17YJCZH228); the Natural Science Foundation of Shandong Province, China (Project No. ZR2017-BD004); the China Postdoctoral Science Foundation (Project No. 2017M612340); the Research project on the cultivation and reform of teaching achievements of Shandong Normal University (2019XM42) and the Shandong Social Science Planning Fund Program (Project No. 19BJCJ23).

**Data Availability Statement:** Data available in a publicly accessible repository that does not issue DOIs Publicly available datasets were analyzed in this study. This data can be found here: http://www.stats.gov.cn/tjsj/tjcbw/202008/t20200826_1785896.html.

**Acknowledgments:** The authors extend great gratitude to the anonymous reviewers and editors for their helpful review and critical comments.

**Conflicts of Interest:** The authors declared no conflicts of interest.

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
