# Peer review of "Recessive Transition Mechanism of Arable Land Use Based on the Perspective of Coupling Coordination of Input–Output: A Case Study of 31 Provinces in China"

_land, doi:10.3390/land10010041_

Round 1

Reviewer 1 Report

The article is very interesting, but I lack an explanation in relation to the natural characteristics. There is a lack of general physical-geographical regionalization of 31 areas. It should be noted that the predominant type of land at 3500 meters above sea level has different input requirements than land in the coastal part of the country. At the same time, the areas in the monsoon part of the country (Hainan) have different requirements than the area in Inner Mongolia.

It is necessary to add to the text the part devoted to the results in relation to these variables. Finding out their relationship will support the results of the work.

Author Response

Response to Reviewer 1 Comments

Point : The article is very interesting, but I lack an explanation in relation to the natural characteristics. There is a lack of general physical-geographical regionalization of 31 areas. It should be noted that the predominant type of land at 3500 meters above sea level has different input requirements than land in the coastal part of the country. At the same time, the areas in the monsoon part of the country (Hainan) have different requirements than the area in Inner Mongolia. 

It is necessary to add to the text the part devoted to the results in relation to these variables. Finding out their relationship will support the results of the work.

Response: Thank you for your insight. China is a vast country, the 31 provinces in this study belongs to different physical geographical districts, and have very different geographical conditions. In the revised discussion (section 4.1, line 462-535, page22-25), we have made deep-going casual analysis of the differences of coupling relation of input-output on arable land by considering the difference of terrain, water, heat, soil and planting structure. The revised section 4.1 can help readers better understand the results of this study by seeing geographical differences of arable land use in China.

Reviewer 2 Report

The main aim of the paper was to explore the law of the recessive transformation mechanism of arable land use based on the input-output perspective. The problem has been correctly formulated and described. The novelty of the paper was also highlighted. The research method developed is appropriate for the purpose of the study and it is described rather clearly. The most important findings from empirical research are clearly described. Discussion included in the fourth section of the paper is interesting, while policy implications of arable land use transition are of operational nature and constitute a real value added of the study.

Below there are some specific comments how to improve the paper.

It would be better not to repeat the words from the title of the paper as keywords.

Line 28: How is the urbanization rate defined? It should be explained.

Line 29-30: How is the proportion of China’s nonagricultural economy measured? Is it share in GDP or any other indicator? It is not clear.

Line 87: “Lv realized…” – the name of the author is not correctly written down.

Line 88: What does the term “sustainable arable land use” mean for authors? This concept should be described in more detail. In this context the issue of input-output relationship on arable land should be introduced and briefly discussed.

Line 96-99: References to previous studies should be given.

Line 123-120: Selection of indicators should be described in more detail and justified from the substantive point of view and following the literature review.

Section 3: As mentioned, research results are clearly described, however, the more extensive application of cause-and-effect analysis would be highly recommended.

Line 446-447: Adequate references to previous studies should be given.

Section 5: Conclusions are consistent with the manuscript content. However, it is also a good practice to indicate the limitation of the research and directions for further research.

As far as formal remarks are concerned, sources of all Figures and Tables should be given. Individual maps within the Figures are too small and it is hard to read them.

Author Response

Reviewers' comments:

The main aim of the paper was to explore the law of the recessive transformation mechanism of arable land use based on the input-output perspective. The problem has been correctly formulated and described. The novelty of the paper was also highlighted. The research method developed is appropriate for the purpose of the study and it is described rather clearly. The most important findings from empirical research are clearly described. Discussion included in the fourth section of the paper is interesting, while policy implications of arable land use transition are of operational nature and constitute a real value added of the study.

Below there are some specific comments how to improve the paper.

Point 1: It would be better not to repeat the words from the title of the paper as keywords.

Response 1: Thank you. We changed the keywords as “recessive transformation of arable land use; input-output; spatio-temporal variation; movement of spatial barycenter; optimization of arable land use” in the revised manuscript.

Point 2: Line 28: How is the urbanization rate defined? It should be explained.

Response 2: Thank you for your remind. It’s necessary to describe the definition of urbanization rate clearly in this manuscript. We have added the detailed definition of the urbanization used in this study in line 26-28, paragraph 1, page 1.

Point 3: Line 29-30: How is the proportion of China’s nonagricultural economy measured? Is it share in GDP or any other indicator? It is not clear.

Response 3: Thank you for your remind. We have added the detailed definition of the China’s nonagricultural economy in this study in line 29-31, paragraph 1, page 1.

Point 4: Line 87: “Lv realized…” – the name of the author is not correctly written down.

Response 4: Thank you. We have corrected the written in line 88.

Point 5: Line 88: What does the term “sustainable arable land use” mean for authors? This concept should be described in more detail. In this context the issue of input-output relationship on arable land should be introduced and briefly discussed.

Response 5: Thank you for your valuable suggestion. We have added more detailed concept and discussion of “sustainable arable land use” in line 88-93, page 2.

Point 6: Line 96-99: References to previous studies should be given.

Response 6: Thank you. We have added relevant references in line102, page 3.

Point 7: Line 123-120: Selection of indicators should be described in more detail and justified from the substantive point of view and following the literature review.

Response 7: Thank you. We have added detailed description of indicator selection in line 127-142, page 3.

Point 8: Section 3: As mentioned, research results are clearly described, however, the more extensive application of cause-and-effect analysis would be highly recommended.

Response 8: Thank you for your insight. China is a vast country, the 31 provinces in this study belongs to different physical geographical districts, and have very different geographical conditions. In the revised discussion (section 4.1, line 462-535, page22-25), we have made deep-going casual analysis of the differences of coupling relation of input-output on arable land by considering the difference of terrain, water, heat, soil and planting structure. The revised section 4.1 can help readers better understand the results of this study by seeing geographical differences of arable land use in China.

Point 9: Line 446-447: Adequate references to previous studies should be given.

Response 9: Thank you. We have added relevant references in the first paragraph of section 4.1, page 22.

Point 10: Section 5: Conclusions are consistent with the manuscript content. However, it is also a good practice to indicate the limitation of the research and directions for further research.

Response 10: Thank you. We have added the limitation of this research and outlook for future research in the last paragraph of the “Conclusion” section.

Point 11: As far as formal remarks are concerned, sources of all Figures and Tables should be given. Individual maps within the Figures are too small and it is hard to read them.

Response 11: Thank you. For better readability, we changed the arrangement of figure2-9 as line 4x2 columns. In the new figures2-9, the size of the figures are enlarged, and the fonts are larger. However, the new figures take up more space of the manuscript. If you and the editor prefer to reduce the size of the figures, we can revise the figures again.

Round 2

Reviewer 2 Report

The paper is much improved and I believe that the changes made have significantly enhanced the quality, clarity and scientific soundness of the manuscript. The paper could be accepted as it is. However, there are three points of technical nature to be made. Sources of all tables and figures should be given, while in lines 443-459 interspace should be adjusted to the editorial requirements. In line 26 space before the bracket with the first reference is missing.

Author Response

The paper is much improved and I believe that the changes made have significantly enhanced the quality, clarity and scientific soundness of the manuscript. The paper could be accepted as it is. However, there are three points of technical nature to be made.

Point 1: Sources of all tables and figures should be given.

Response 1: Thank you. We have uploaded eps format images and XLSX format tables to the submission system.

Point 2: In lines 443-459 interspace should be adjusted to the editorial requirements.

Response 2: Thank you. We have adjusted the interspace as the editorial requirements in line 443-459, page 22.

Point 3: In line 26 space before the bracket with the first reference is missing.

Response 3: Thank you for your reminding. We have added spaces before the brackets to all references.
